# Advanced Skeletal Ossification Is Associated with Genetic Variants in Chronologically Young Beef Heifers

**DOI:** 10.3390/genes14081629

**Published:** 2023-08-15

**Authors:** Katie A. Shira, Brenda M. Murdoch, Kimberly M. Davenport, Gabrielle M. Becker, Shangqian Xie, Antonetta M. Colacchio, Phillip D. Bass, Michael J. Colle, Gordon K. Murdoch

**Affiliations:** 1Department of Animal, Veterinary and Food Sciences, University of Idaho, Moscow, ID 83844, USA; 2Department of Animal Sciences, Washington State University, Pullman, WA 99164, USA

**Keywords:** beef heifers, advanced skeletal maturity, bone

## Abstract

Osteogenesis is a developmental process critical for structural support and the establishment of a dynamic reservoir for calcium and phosphorus. Changes in livestock breeding over the past 100 years have resulted in earlier bone development and increased physical size of cattle. Advanced skeletal maturity is now commonly observed at harvest, with heifers displaying more mature bone than is expected at 30 months of age (MOA). We surmise that selection for growth traits and earlier reproductive maturity resulted in co-selection for accelerated skeletal ossification. This study examines the relationship of single nucleotide polymorphisms (SNPs) in 793 beef heifers under 30 MOA with USDA-graded skeletal maturity phenotypes (A-, B-, C- skeletal maturity). Further, the estrogen content of FDA-approved hormonal implants provided to heifers prior to harvest was evaluated in association with the identified SNPs and maturities. Association tests were performed, and the impact of the implants were evaluated as covariates against genotypes using a logistic regression model. SNPs from the *ESR1*, *ALPL, PPARGC1B*, *SORCS1* genes, and SNPs near *KLF14*, *ANKRD61*, *USP42*, *H1C1*, *OVCA2*, microRNA *mir-29a* were determined to be associated with the advanced skeletal ossification phenotype in heifers. Higher dosage estrogen implants increased skeletal maturity in heifers with certain SNP genotypes.

## 1. Introduction

Bone maturation is an essential component of mammalian growth and is important in preparing an animal for maturity [1,2,3]. There has been considerable selection pressure on beef heifers to mature earlier in life and produce a calf to maintain profitability for producers [1,2,3,4]. Bone maturation and reproductive maturation happen concurrently as mammals go through puberty. As animals reach earlier reproductive maturity, their skeletal system also undergoes advanced maturity. The accelerated maturation of bone in chronologically young cattle is referred to as advanced skeletal ossification [5].

Until recently, bone maturation affected the compensation provided to producers when their livestock were harvested [6]. Prior to 2017, carcasses were assessed for their maturity based on the degree of skeletal ossification, which was used to establish their months of age (MOA). They were classified as A- (9–30 MOA), B- (30–42 MOA), C- (42–72 MOA), D- (72–96 MOA) and E- (>96 MOA) maturity [7]. It was identified that cattle, especially heifers, known to be under 30 MOA (A- maturity) were being mislabeled as over 30 MOA (B–E maturity) when evaluating the degree of skeletal ossification at harvest [8]. Some animals exhibit advanced bone physiological maturation, and therefore, their degree of ossification does not correspond with their chronological age. Cattle under 30 MOA based on dentition are currently preferred as the financial compensation to producers is greater. There is an expense associated with handling a carcass for specified risk material when the animal is over 30 MOA [9]. Further, cattle under 30 MOA have more preferred quality attributes as chronologically younger cattle often exhibit improved tenderness and marbling and more acceptable color [10,11,12]. Previous studies evaluated how advanced skeletal maturity affects meat quality and reported that advanced skeletal ossification in chronologically young cattle does not compromise quality [13,14,15]. The grading system was revised in 2017 when the beef industry changed to use dentition or birth records for assessing age [16]. These grading system changes helped mitigate the financial losses to producers when animals are graded as advanced maturity when they are under 30 MOA, but the underlying physiology of the animal has not changed. Advanced skeletal ossification is still occurring in the cattle industry and is a phenotype worthy of investigation. Bone maturation is a complex process regulated by multiple genes and physiological pathways [17,18]. Bone development and skeletal ossification are critical for the dynamic regulation of minerals, and understanding the genetic and physiological mechanisms involved in advanced maturity will provide valuable insights into this important aspect of cattle production [19].

In this study, heifer carcasses of A-, B-, and C- skeletal maturity that were chronologically young (<30 MOA) as identified by dentition were selected for allelic discrimination and sequencing analyses. This study evaluates genetic variants in and near physiologically relevant genes for association with skeletal maturity and validates previously reported variants [20]. The results of these analyses improve the current understanding of candidate genes that may be involved in the advanced maturity phenotype and suggest genetic markers that can be used to predict genetic propensity for advanced skeletal ossification.

## 2. Materials and Methods

### 2.1. Sample Collection

The USDA grading system used prior to 2017 was employed in this study to classify heifers as either A-, B-, or C- skeletal maturity. Muscle or ear notch samples were collected from USDA-inspected mixed breed beef heifers confirmed to be under 30 MOA by dentition. Sampling was conducted to be representative of the average composition of the animals processed through the Pacific Northwest beef industry. Sample collection occurred at Washington Beef in Toppenish, Washington between the dates of 10 July 2019 and 10 January 2020. There were 15 different collection days within this time frame. The same two USDA graders identified the skeletal maturity grade for each carcass collected for the duration of the study. All samples were collected and placed on ice and transported back to the University of Idaho located in Moscow, Idaho. From the heifers collected, 545 were classified as A- skeletal maturity, 249 heifers were classified as B- skeletal maturity, and 244 heifers were classified as C- skeletal maturity. In an effort to make sample sizes more consistent for statistical analyses, 300 A- maturity heifers were carried forward with the B- and C- maturity samples for allelic discrimination and sequencing. All tissue samples were stored at −20 °C until DNA isolation.

### 2.2. Cattle Record Collection

At the time of sample collection, feedlot names and lot numbers for each group of cattle were provided from the harvesting facility. A total of 21 feedlots were included in the study and were contacted to obtain cattle records. The primary goal was to obtain hormonal anabolic implant information. Implant information was successfully gathered for *n* = 578 of the heifers sampled.

### 2.3. DNA Isolation and Quantification

DNA isolation from tissue samples was carried out using the phenol–chloroform method as previously described and eluted in nuclease-free water [21]. Quantification of DNA was performed using the Nanodrop Spectrophotometer (ThermoFisher Scientific, Waltham, MA, USA) and/or AccuBlue^®^ Broad Range dsDNA Quantitation (Biotium, San Francisco, CA, USA) methods. Nanodrop Spectrophotometer methods followed DNA quantity and quality parameters as previously described [22], and AccuBlue dsDNA quantification was carried out using the AccuBlue^®^ Broad Range dsDNA Quantitation Kit with DNA standards following the manufacturer’s protocol.

### 2.4. Allelic Discrimination and Sequencing

Genotyping of candidate estrogen receptor 1 (*ESR1*) and alkaline phosphatase (*ALPL*) SNPs were validated in heifers (*n* = 793). This was performed with custom allelic discrimination assays that were designed using Primer Express™ 3 v3.0.1 (Life Technologies, Corp., ThermoFisher Scientific, Waltham, MA, USA). Primers and probes for the *ESR1* gene were designed around two candidate SNPs (Appendix A). The primer and probe sequences were entered into the ThermoFisher Scientific website to generate TaqMan^®^ Custom SNP non-human Assays. These assays were used in combination with the TaqPath™ ProAmp™ Master Mix (ThermoFisher Scientific, Waltham, MA, USA). The reaction for each animal was run on the Life Technologies ViiA7 Real-Time PCR System (Applied Biosystems) as an endpoint reaction with the following conditions: pre-read 60 °C for 30 s, initial denature at 95 °C for 5 min, denature at 95 °C and anneal/extend at 60 °C for 40 cycles, and post-read is set to 60 °C for 30 s and then set to hold at 4 °C.

Thirteen candidate regions (Appendix A) were selected for targeted sequencing with 634 heifers. These candidate regions were selected for the presence of genes with known or suspected roles in bone growth and development [23,24,25,26,27,28]. Primer pairs for short-read sequencing were designed using Primer Plus 3 software and gene-specific 250 base pair sequences obtained from the NCBI Bos taurus genome ARS-UCD1.2 [29]. PCR amplification was performed according to recommended instructions, and amplicon approximate sizes were confirmed using gel electrophoresis of each location for a subset of samples and then subsequently spot-checked. Amplicons were quantified using the AccuBlue^®^ Broad Range dsDNA Quantitation (Biotium, San Francisco, CA, USA) methods to obtain estimates of the DNA amplicon concentrations. Amplicon PCR was performed using FastStart™ High Fidelity PCR System, dNTPack (Roche Sequencing Solutions, Inc., Pleasanton, CA, USA) with the custom primer pairs. Amplicons (10 µL of each reaction) were then purified using the HighPrep™ PCR bead cleanup according to the manufacturer’s instructions (MagBio Genomics Inc., Gaithersburg, MD, USA). The purified amplicons were subsequently indexed using the Nextera XT Index Kit v2 primers (Illumina Inc., San Diego, CA, USA), as per the manufacturer’s protocol. Post indexing, 10 µL of each sample was again subjected to bead cleanup, as described above. Samples were pooled into a microcentrifuge tube as recommended by the library preparation kit, and sequencing was completed on the iSeq100 System (Illumina Inc., San Diego, CA, USA).

### 2.5. Sequence Mapping and Variant Calling

The paired-end reads of 634 samples were controlled and trimmed using FastQC v0.11.3 and Trimmomatic 0.4 [30]. The clean sequences were aligned to the reference genome ARS-UCD1.2 (GCF_002263795.1) by using the ‘mem’ of bwa v0.7.17-r1188 with default parameters [31]. The variants from each sample were separately called by the HaplotypeCaller model of The Genome Analysis Toolkit (GATK) v4.1.7.0 [32] and then combined to generate the final VCF file of all samples using the GenotypeGVCFs model of GATK with ‘-stand-call-conf 10′. The integration of variation position and gene annotation were extracted using BEDTools v2.26.0 [33].

### 2.6. Genetic Association Analyses

As mentioned previously, the initial sample size of heifers that went into this study is *n* = 793. That total group of 793 heifers was tested with allelic discrimination assays. From that population of 793 heifers, a subset was used for sequencing (*n* = 634). Due to sequencing constraints, the total of 793 heifers could not be used for this portion of the study, and this subset of heifers was selected for having implant information available, and then the remainder were selected to ensure there was a representative number of heifers in each developmental maturity category. The sample size changes from the initial heifers tested in the study due to sample failure and quality control measures. In the allelic discrimination assays, all SNPs had a call rate of ≥98%. The *ESR1* SNP 9:9001509 and *ALPL* SNP 2:131837201 had a call rate of 99.1% (*n* = 786 heifers) and the *ESR1* SNP 9:90115650 had a call rate of 98.0% (*n* = 777 heifers). Quality control of sequencing data was accomplished first through sample call rate and then through marker quality thresholds. Of the 634 heifers successfully sequenced, those with calls at less than 50% of identified SNPs were removed from analyses, resulting in a dataset of *n* = 466 heifers. The SNPs identified through sequencing were filtered to remove those with a minor allele frequency <1%, Hardy–Weinberg Equilibrium *p*-value < 1 × 10^−6^ and marker call rates <70%, resulting in a dataset of 49 SNPs carried forward for association and regression analyses. Additionally, *n* = 349 of the heifers passing quality control had known estrogen implant data. As described previously, the exogenous estrogen concentration administered to each heifer was identified from information provided by the feedlots. These samples were further analyzed in a logistic regression analysis to evaluate the effect of SNP genotypes while accounting for exogenous estrogen exposure as a fixed effect.

The phenotypic associations of SNPs identified through allelic discrimination and sequencing assays were evaluated using genetic association analyses in SNP and Variation Suite™ v8.9.0 (Golden Helix, Inc., Bozeman, MT, USA, www.goldenhelix.com accessed on 5 July 2023). In addition to the individual association tests, logistic regression was used to evaluate heifers with estrogen concentration as a covariate. The estrogen concentration was determined from implant information provided by producers to the feedlots. Due to the lack of existing data on the mode of inheritance for this phenotype, association models were evaluated in Allelic, Basic, and Genotypic models, and both association and logistic regression models were evaluated in an Additive, Dominant, and Recessive inheritance in order to identify the model of best fit. Models with the most robust *p*-values were carried forward for final analyses.

The effect size of each SNP was calculated from contingency tables using Cramer’s V statistic [34]. This statistic is reported from 0, which indicates no relationship, to 1, which indicates a perfect relationship. Cramer’s V was calculated in R v. 4.2.3 with the package rcompanion [35]. Statistical significance was determined at *p* < 0.05, and trends were determined between 0.05 and 0.10.

## 3. Results

### 3.1. Association Analyses

Using custom-designed TaqMan™ allelic discrimination assays, the associations of three SNPs with advanced skeletal maturity were validated in heifers. Two SNPs within the gene *ESR1* and one SNP within the gene *ALPL* (Table 1) were found to be associated with advanced skeletal ossification in beef heifers at harvest through association and regression analyses. *ESR1* SNP_9:90115650 was significantly associated (*p* = 0.006) with advanced skeletal ossification in beef heifers when using a dominant model to test for association. Data indicates that when heifers have one or more copies of the G allele, they have normal ossification and with two copies of the A allele, they grade as a C- maturity carcass at a higher frequency (Figure 1A). The second SNP in *ESR1,* SNP_9:90015095, was not found to be significantly associated with the A- vs. B- vs. C-maturity phenotype (*p* = 0.220). A trend (*p* = 0.074) was identified with *ALPL* SNP_2:131837201 when using the recessive association model. At this location, when heifers have the AA genotype, they were less likely to grade as a C- maturity carcass, which represents the most advanced maturity out of the heifers included in the study.

The targeted sequencing analysis also identified significant SNPs associated with advanced carcass maturity in beef heifers. Seventeen SNPs were identified from targeted sequencing of *GHR*, *PTHLH*, *RUNX2* and *SP7*; however, after testing, these markers were not significantly associated with advanced maturity (Appendix A). Additional SNPs were identified from sequenced regions that shared similarities with targets. These eight SNPs were found to be significantly associated or showed statistical trends for association with beef heifer carcass maturity (Table 1 and Table 2). SNPs were either intronic or positioned within a ± 100 kb region upstream or downstream of genes (Table 3).

Testing heifers as A- vs. B- vs. C- maturity identified four significant SNPs. SNP_7:61,142,080 (*p* = 0.006) and SNP_7:61,142,117 (*p* = 0.0007) are both in the *PPARGC1B* gene; intergenic SNP, SNP_10:76,769,110 (*p* = 0.030) is 68 kb upstream of the *PLEKHG3* gene and 58 kb downstream of the *PPPP1R36* gene; intergenic SNP_25:38,081,982 (*p* = 0.002) is 5 kb upstream *USP42* and 16 kb downstream of the *ANKRD61* gene. As advanced skeletal maturity increased from A- to B- to C-, the alternate allele frequency also increased for SNP_7:61,142,080, SNP_7:61,142,117 (Figure 1B), and 25:38,081,982. Inversely SNP_10:76,769,110 had the opposite where the alternate allele frequency was higher in the A- maturity (47%) and decreased in frequency with B- (40%) and C- (39%) maturity.

As presented in Table 4, when testing A- vs. B- and C- maturity, three SNPs were identified as significant. These SNPs were SNP_19:23,066,541 (*p* = 0.045) found 6 kb upstream of the *H1C1* gene and 665 kb downstream from the *OVCA2* gene, SNP_26:28,004,085 (*p* = 0.012), which is within the *SORCS1* gene, and SNP_10:76,769,055 (*p* = 0.008) that is 68 kb upstream the *PLEKHG3* gene and 58 kb downstream of the *PPPP1R36* gene. The SNP_19:23,066,541 has a higher alternate allele frequency associated with the B- and C- maturity (4.1%) compared to the A- maturity (1.7%). The SNP 10:76,769,055 also has a higher frequency of alternate alleles (15.5%) for the B- and C- maturity, compared to the A- maturity (9.4%). The opposite was seen with SNP_26:28,004,085; for this SNP, A- maturity had a higher alternate allele frequency (4%) than the B- and C- maturity (1.3%).

Identifying SNPs associated with C- maturity could facilitate understanding of and selection against this undesired phenotype. The SNP_4:94,584,161 was found to be significant (*p* = 0.032) when testing C- vs. A- and B- maturity. This SNP is 63 kb downstream of the *KLF14* gene, and 52 kb upstream of *mir-29a*. There is a higher frequency (2.6%) of the alternate allele in C- maturity heifers compared to the frequency (0.8%) in A- and B- maturity heifers.

### 3.2. Regression Analyses

A genotypic logistic regression model was implemented in the SNP validation to identify whether the estrogen concentration provided in the pre-harvest finishing phase further influenced advanced skeletal ossification. There were two different concentrations of estrogen provided to heifers by FDA-approved commercial implants: 20 mg and 28 mg (Table 4). In the allelic discrimination analyses, the *ESR1* SNP_9:90115650 (*p* = 7.54 × 10^−19^) and SNP_9:90015095 (*p* = 4.07 × 10^−15^) were associated with advanced skeletal ossification as well as *ALPL* SNP 2:131837201 (*p* = 0.002) in heifers with a 28 mg estrogenic implant compared to heifers that received a 20 mg implant (Table 5). Through this analysis, it was determined that accounting for the estrogen concentration of the implant as a covariate greatly improved the power of the three SNPs, especially the *ESR1* targets.

Logistic regression analyses were further utilized to identify if the estrogen concentration of implants increases the chances of a heifer grading as advanced maturity at SNPs identified through sequencing (Table 5). Six of the SNPs previously described were significant when including the estrogen concentration as a covariate. The markers SNP_7:61,142,117 (*p* = 0.045) and SNP_25:38,081,982 (*p* = 0.012) were significant when using the dominant model testing heifers as A- vs. B- vs. C- maturity, and SNP_10:76,769,110 was significant (*p* = 0.009) when using an additive model to test heifers as A- vs. B- vs. C- maturity. When heifers were tested as A- vs. B- and C- maturity in an additive model, three significant SNPs were identified, including SNP_10:76,769,055 (*p* = 0.004), SNP_19:23,066,541 (*p* = 0.036), and SNP_26:28,004,085 (*p* = 0.009). These results show that accounting for estrogen improved the power and significance of SNP_10:76,769,110, SNP_10:76,769,055, SNP_19:23,066,541, SNP_26:28,004,085 shown in Table 5, while using estrogen as a covariant did not improve the significance of the other SNPs.

## 4. Discussion

### 4.1. SNP Validation

This study validated SNPs from a pilot study [20] and further identified novel SNPs associated with advanced skeletal ossification phenotypes in chronologically young beef heifers. The issue of advanced skeletal ossification is relevant to beef heifer production as bone tissue serves as a dynamic storage depot for minerals that are essential to growth, development, reproduction, and production [36,37]. There is evidence that some of the genes evaluated in this study influence bone growth, carcass weight, fat deposition, mineral balance, and sexual maturity of females [20,38,39,40]. Many of these genes have a strong biological link to bone growth or regulation of cells known to influence osteogenesis.

Logistic Regression analyses were used to evaluate implanted estrogen as a covariate in this study. The results of the regression analyses with estrogen concentration were significant, but it is important to consider the relatively small sample size that is driving the significance. While there was a smaller population receiving a higher estrogen-containing implant (28 mg) compared to the rest of the population, which received 20 mg of estrogen in their finishing phase, we remain confident that the identified associations likely represent an environmental and genotypic interaction resulting in a higher probability of advanced maturity. These heifers were raised across different locations in the pacific northwest United States, so implant information was not received from earlier stages in these heifers’ lives; only information from the feedlots was received and included in our analyses. Further work is needed in order to more fully understand the relationship between exogenous estrogen exposure and advanced maturity grading.

The SNP within *ESR1* that was previously identified in a pilot study was found to be significantly associated with advanced skeletal ossification in this larger population of beef heifers. Beef heifers that had a G allele in this location had a normal rate of bone growth, but heifers with two copies of the A allele were more likely to present the phenotype of advanced skeletal ossification. The *ESR1* gene encodes proteins that not only regulate many reproductive functions in multiple species but also contribute to the regulation of bone homeostasis and appendicular bone growth in females, as demonstrated by mouse studies [41,42,43]. The rate of bone growth and ossification is accelerated as females reach puberty due to an increase in estrogen production and free serum estrogen [44,45]. This suggests that when a heifer with at least one G allele in this location reaches maturity, her rate of ossification may proceed at a physiologically normal rate compared to her counterparts with a homozygous AA or heterozygous AG genotype. This could be due to the rise in estrogen production that is caused by the onset of puberty; however, this study did not analyze the age of puberty or estrogen levels. When estrogen increases, it prolongs the osteoblast lifespan through the action of decreasing osteoblast apoptosis. This allows for more active bone cells during osteogenesis [46,47]. For this reason, implant information was obtained and included as a covariate in this experiment. It is known that estrogen not only helps with muscle growth, but also bone maintenance and deposition through the action of traditional estrogen receptor signaling pathways [48,49,50].

The previously reported SNP within *ALPL* from Colacchio and colleagues [20] was validated in this study. The regression analysis found that this genotype was associated with advanced maturity, and logistic regression analysis suggests that skeletal maturity is impacted by the estrogen provided to heifers in implants. These observations align with the known biology of alkaline phosphatase, an enzyme that is regulated by *ALPL* and is responsible for bone mineralization [51,52]. In human studies, alkaline phosphatase was found in the cell membranes of hypertrophic chondrocytes and osteoblasts, which occurs in the zone of ossification [51]. Under estrogen exposure, proliferating chondrocytes decrease in activity, and osteoblasts increase their anabolic activity [46,53], which further supports why estrogen levels in implants are important to evaluate. The increased mineralization could be an indirect effect of estrogen slowing down chondrogenesis (cartilage formation) by decreasing proliferating chondrocytes. This would increase hypertrophic chondrocytes and increase osteogenesis by increasing osteoblast activity and mineralization.

### 4.2. Targeted Sequencing

Heifers that had the alternate allele for SNP_7:61,142,080 and SNP_7:61,142,117 in the *PPARGC1B* gene were graded as B- and C- maturity at a higher frequency. *PPARGC1B* plays a critical role in mitochondrial biogenesis, as identified in cancer studies [54]. Knock-down studies indicate that this gene inhibits osteoclast differentiation and mitochondrial biogenesis in mice [55]. Osteoclasts play an important role in endochondral ossification and homeostasis in the skeletal system. Osteoclasts and vasculature invade the cartilage simultaneously to prepare the cartilage for ossification [56,57]. With this gene being linked to osteoclast differentiation and an increase in mitochondria, physiologically, it is plausible that these two processes are active and influence the ossification rate. This may lead to the ossification process occurring more rapidly in the growth plate. Additionally, *PPARGC1B* encodes for PGC-1α, which enhances the activity of nuclear receptors. PGC-1-related coactivator is part of the PGC-1 family and functions similarly. PGC-1-related protein is a coactivator of estrogen receptor 1 [58]. As mentioned previously, *ESR1* is a gene of interest in this study as it encodes for estrogen receptor which plays a role in growth and development.

The SNP_4:94584161 downstream of the *KLF14* gene and upstream of microRNA *mir-29a* was associated with the alternative allele in C- maturity heifers. It is possible that the region of this SNP could be in linkage disequilibrium with a variant that influences the rate of bone growth. The gene *KLF14* is expressed in human bone marrow mesenchymal stem cells, and it is downregulated during osteogenic differentiation, which results in osteoblast production [59]. In human and mouse studies, *KLF14* was identified to be an important regulator of adipocyte development, was expressed at higher levels in females, and its expression changes with age [60,61,62]. The microRNA *mir-29a* is a key regulator of osteogenic differentiation. Previous studies identify that the expression of *mir-29a* increases extracellular matrix mineralization through Wnt signaling, promotes angiogenesis and osteogenesis, and promotes osteoclastogenesis, as identified in mouse research [63,64,65]. The fact that these genes can regulate both adipose and bone is relevant to the stage of life that these heifers were harvested at. Cattle are typically fed high-energy diets during finishing to obtain certain levels of fat deposition (marbling) pre-harvest. Over time, cattle intended for harvest have been gradually selected for more efficient growth and increased marbling. Application of selection pressures for marbling and growth may have been unintentionally selecting for the alternate allele at SNP_4:94584161, which could result in bone mineralization occurring quicker.

The alternate allele SNP_26:28004085 positioned within the *SORCS1* gene was associated with A- maturity heifers. When heifers are both homozygous for the reference T allele and have a higher level of estrogen, they are more likely to grade as B- and C- maturity carcasses. The genotype and phenotype at this SNP were highly correlated with an effect size of 0.974. In previous cattle studies, *SORCS1* was associated with increased carcass weight and linked to carcass fatness [66,67]. Carcass weight, fat thickness, and marbling are important traits that have been studied and selected for in the beef industry as they relate to meat quality [68,69,70,71]. Over time, the selection pressure for some of these traits may have also unintentionally increased the frequency of the alternate allele for *SORCS1*. Bone has to ossify to support increased carcass weight as an animal reaches maturity; however, it has been shown that increasing the load bearing will also increase the rate of bone ossification [72]. Biological evidence supports this SNP being associated with bone growth, particularly as it relates to increased carcass weight.

The SNP_10:76769110 is downstream of the gene *PPP1R36.* This gene is associated with the regulation of spermatogenesis in males, which is developmentally important, and has been identified to be upregulated in bovine embryos from high fertility sires in comparison to embryos of low fertility sires [73,74]. *PPP1R36* promotes and enhances autophagy during spermatogenesis [74]. This explains its role in male reproduction, and while that was not the population tested in this study, the upregulation of *PPP1R36* in embryos of high fertility sires suggests that this gene could be affecting the female offspring. This study links *PPP1R36* to the maturity of bone in females and proposes this gene for further investigation to understand its role in female livestock.

The SNP_ 25:38,081,982 is near the genes *USP42* and *ANKRD61.* The USP family plays an empirical role in posttranslational modification and protein homeostasis by regulating cellular activity [75]. USP proteins are involved in aspects of bone growth as identified in humans and mice [76,77,78,79,80,81]. *USP42* has been shown to play very important roles in transcription, where it functions to deubiquitylate histones to regulate transcription. Genes *USP42* and *RUNX1* are both studied in leukemia for their involvement in chromosomal translocations. *RUNX1* in healthy tissue encodes proteins that regulate hematopoiesis and is suggested to play a role in embryogenesis and spermatogenesis, as reported in mice [82,83,84,85]. Whereas the gene *ANKRD61* encodes for ankyrin repeat domain 61 protein, and ankyrin repeat proteins can support USP and regulate protein-to-protein interactions [86]. Both genes influence or regulate ubiquitylation. Ubiquitin is a regulatory protein important in cell signaling [75,76], including bone cells, as ubiquitin-dependent endocytosis is required to remove the receptors on the surface. This includes receptors that bind bone morphogenic proteins (BMP). Additionally, it can inhibit the Smad pathway, which upregulates osteoblast-specific genes [77,78,79]. BMP signals begin with the expression of SOX9, which promotes mesenchymal stem cells to differentiate into chondrocytes [17,87]. BMPs play a key role in osteogenesis, as noted in a study on mice where the loss of BMP-2 and BMP-4 impaired osteogenesis as BMPs can signal for osteoblast differentiation [88,89]. Although this is evidence of how ubiquitin regulates growth in bone, it is also important to other aspects of growth across the body. The SNP identified in this study is between genes, but these genes are important to cell signaling, as listed. The presence of the alternate SNP_ 25:38,081,982 allele could be involved with the regulation of either or both above-mentioned genes and therefore related to the rate of growth that is seen in heifers that experience advanced maturity.

The SNP 19:23,066,541 is another intergenic SNP, but the genes located near this SNP are not as well studied as other genes discussed in this paper. *H1C1* and *OVCA2* are the closest upstream and downstream genes to this location. *H1C1* is a tumor suppressor gene and is primarily studied in human cancer research [90], and *OVCA2* has been mostly studied for its expression in breast and ovarian tumors in humans [91,92], but there is little evidence of this gene’s role in cattle.

This research identified genetic markers that are associated with advanced skeletal ossification as determined by skeletal maturity grade in young beef heifers at the time of harvest. Many of these markers are within, or very near genes that are important to beef production and growth, as outlined in this manuscript. Considering that the population in this study was comprised of post-pubertal beef heifers and the links that some of these markers have to reproduction, it would be insightful to evaluate whether these SNPs are also associated with heifer fertility. Future work is necessary to evaluate potential connections between these markers and precocious puberty and/or fertility rates in young heifers. As reported by others, the rate at which a heifer reaches puberty and breeds for the first time will affect her full production potential across her lifetime [30]. This can impact the efficiency and profitability of a livestock operation. It would be justifiable to design a follow-up study to implicitly assess the presence of the SNPs we reported in the context of thorough phenotypic assessment of reproductive traits in heifers and cows. Not only would it be beneficial to evaluate these SNPs further for reproduction, but it would also be informative to evaluate the rations provided to heifers when investigating these SNPs. This study attempted to incorporate this aspect, but not enough data were obtained to do a full analysis. Other data show that the quality of the ration does affect heifer sexual maturity [93,94], so we hypothesize that the quality of ration a heifer is on may also affect skeletal maturity.

## 5. Conclusions

Significant SNPs associated with advanced skeletal ossification of chronologically young beef heifers were identified in the *ESR1*, *ALPL, PPARGC1B*, *SORCS1* genes, and near *KLF14*, *ANKRD61*, *USP42*, *H1C1*, *OVCA2*, and microRNA *mir-29a*. These SNPs were identified in beef heifers that were under 30 MOA at the time of harvest, as confirmed via dentition. This study identified biological support which may link these SNPs to earlier ossification. Furthermore, this study has provided evidence that higher levels of exogenous estrogen supplied to beef heifers as components of FDA-approved implants during their finishing phase are more susceptible to advanced ossification. Producers with heifers that have these genotypes could potentially mitigate the issue of advanced ossification by utilizing a lower dose estrogen implant to reduce the likelihood of them presenting with advanced skeletal ossification at harvest.

## Figures and Tables

**Figure 1 genes-14-01629-f001:**
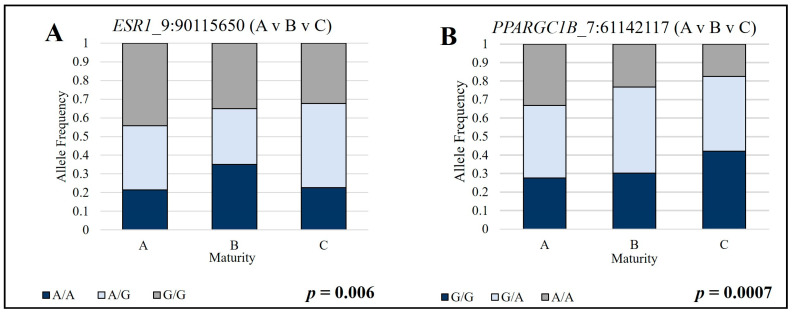
The genotype frequency for each maturity category of A-, B- and C- skeletal maturity across SNPs tested for association with advanced skeletal ossification. (**A**) A- maturity heifers have the highest frequency of G alleles, while C- maturity has the highest frequency of A alleles. (**B**) A maturity heifers have the highest frequency of T alleles, B maturity have a higher frequency of T alleles than C maturity and C alleles than A maturity, and C maturity have the highest frequency of C alleles.

**Table 1 genes-14-01629-t001:** Results of allelic association tests for carcass maturity classification. * Signifies results from the allelic discrimination test. Genomic elements located within ±100 kb of each SNP are noted, positions are given for the ARS-UCD1.3 genome assembly. Effect size was calculated using Cramer’s V.

Position Chr:BP	Ref./Alt.	Test	*p*-Value	Effect Size	Alt Allele Frequency
4:94,584,161	A/G	C vs. AB	0.032	0.099	0.026|0.008
7:61,142,080	T/C	A vs. B vs. C	0.006	0.107	0.750|0.805|0.847
7:61,142,117	A/G	A vs. B vs. C	0.0007	0.128	0.472|0.535|0.623
10:76,769,055	G/A	A vs. BC	0.008	0.489	0.094|0.155
10:76,769,110	G/A	A vs. B vs. C	0.030	0.086	0.472|0.400|0.390
19:23,066,541	C/T	A vs. BC	0.045	0.101	0.017|0.041
25:38,081,982	G/C	A vs. B vs. C	0.002	0.104	0.000|0.005|0.025
26:28,004,085	T/C	A vs. BC	0.012	0.974	0.040|0.013
* 9:90115650	A/G	A vs. B vs. C	0.006	0.087	0.740|0.683|0.705
* 9:90015095	C/T	A vs. B vs. C	0.220	0.090	0.665|0.670|0.722
* 2:131837201	G/A	A vs. B vs. C	0.074	0.092	0.769|0.783|0.577

**Table 2 genes-14-01629-t002:** Details for SNPs that showed a statistical trend but were not significant in analyses. Genomic elements located within ±100 kb of each SNP are noted, positions are given for the ARS-UCD1.3 genome assembly.

Position Chr:BP	Ref./Alt.	Test	Model	Inheritance	*p*-Value	Genomic Context (±100 kb)
4:15,158,192	T/G	A vs. BC	Logistic Regression	Dominant	0.0634	Intronic *ASNS*
5:38,540,781	A/G	A vs. BC	Association	Recessive	0.0745	Within *YAF2* 3′ UTR, upstream of *GXYLT1*; downstream of *LOC785294*
9:84,791,505	C/T	A vs. BC	Association	Recessive	0.0997	Upstream of *SAMD5*, *LOC100850276*; downstream of *LOC512881*, *LOC101907062*, *LOC101906979*
9:94,978,531	T/C	A vs. BC	Association	Dominant	0.0920	Upstream of *TMEM181*, *LOC101903438*, *SYTL3*; downstream of *TULP4*, *LOC104969639*, *DYNLT1*
11:102,238,700	G/C	A vs. BC	Allelic Association		0.0838	Upstream of *NTNG2*, *MED27*; downstream of *SETX*
11:102,238,757	C/T	C vs. AB	Logistic Regression	Dominant	0.0618	Upstream of *NTNG2*, *MED27*; downstream of *SETX*
12:15,818,603	C/T	A vs. BC	Logistic Regression	Dominant	0.0967	Upstream of *ERICH6B*, *LOC101902486*; downstream of *SIAH3*, *CBY2*
14:23,564,180	A/G	A vs. B vs. C	Logistic Regression	Dominant	0.0563	Upstream of *PENK*, *SDR16C6*
19:23,346,922	G/C	C vs. AB	Association Test	Recessive	0.0908	Upstream of *LOC112442622*, *TSR1*, *LOC112442829*, *LOC112442828*, *SMG6*; downstream of *SGSM2*, *MNT*, *METTL16*, *SRR*
20:63,542,232	A/G	A vs. BC	Logistic Regression	Recessive	0.0837	Upstream of *TAS2R1*; downstream of *LOC104975290*
20:63,542,246	T/C	A vs. B vs. C	Logistic Regression	Recessive	0.0557	Upstream of *TAS2R1*; downstream of *LOC104975290*
22:41,302,660	C/T	C vs. AB	Genotypic Association		0.0987	Intronic *FHIT*

**Table 3 genes-14-01629-t003:** Genomic context of significant SNPs. Genomic elements located within ± 100 kb of each SNP are noted, positions are given for the ARS-UCD1.3 genome assembly.

Position Chr:BP	Genomic Context (±100 kb)
4:94,584,161	Upstream of *KLF14*, *LOC112446482, LOC107132431;* downstream of *LOC107131356*, *SNORA13*, *mir-29a*, *mir-29b-1*
7:61,142,080	Intronic *PPARGC1B;* downstream of *PDE6A, mir-378-1*
7:61,142,117	Intronic *PPARGC1B;* downstream of *PDE6A, mir-378-1*
10:76,769,055	Upstream of *PLEKHG3*; downstream of *LOC104973208*, *PPP1R36*, *HSPA2*, *ZBTB1*
10:76,769,110	Upstream of *PLEKHG3*; downstream of *LOC104973208*, *PPP1R36*, *HSPA2*, *ZBTB1*
19:23,066,541	Upstream of *HIC1*, *RTN4RL1*, *LOC112442776*, *LOC112442621*; downstream of *DPH1*, *OVCA2*, *mir-132*, *mir-212*, *SMG6*
25:38,081,982	Upstream of *USP42*, *EIF2AK1*, *PMS2*, *RSPH10B*; downstream of *ANKRD61*, *AIMP2*, *CYTH3*
26:28,004,085	Intronic *SORCS1*

**Table 4 genes-14-01629-t004:** Hormonal FDA-approved implants were identified to be provided to heifers during the finishing phase prior to harvest in this study. Each implant has the hormone composition listed, and then how many heifers had received each type of implant.

Brand	Estrogen	Testosterone	N Receiving 20 or 28 mg Estrogen
Component TE-200 with Tylan and Revlor XH	20 mg estradiol	200 mg trenbolone acetate	*n* = 802
Synovex Plus	28 mg estradiol benzoate	200 mg trenbolone acetate	*n* = 18

**Table 5 genes-14-01629-t005:** Results of logistic regression analyses for carcass maturity classification. * Signifies results from the allelic discrimination test. In all regression models, estrogen implant concentration was included as a fixed effect. Genomic elements located within ± 100 kb of each SNP are noted, positions are given for the ARS-UCD1.3 genome assembly.

Position Chr:BP	Ref./Alt.	Test	Inheritance	*p*-Value	Alt Allele Frequency
7:61,142,117	A/G	A vs. B vs. C	Dominant	0.045	0.472|0.529|0.571
10:76,769,055	G/A	A vs. BC	Additive	0.004	0.094|0.160
10:76,769,110	G/A	A vs. B vs. C	Additive	0.009	0.472|0.404|0.370
19:23,066,541	C/T	A vs. BC	Additive	0.036	0.017|0.045
25:38,081,982	G/C	A vs. B vs. C	Dominant	0.012	0.000|0.005|0.038
26:28,004,085	T/C	A vs. BC	Additive	0.009	0.040|0.010
* 9:90115650	A/G	A vs. B vs. C	Genotypic	7.54 × 10^−19^	0.740|0.683|0.705
* 9:90015095	C/T	A vs. B vs. C	Genotypic	4.07 × 10^−15^	0.665|0.670|0.722
* 2:131837201	G/A	A vs. B vs. C	Genotypic	0.002	0.769|0.783|0.577

## Data Availability

Data can be provided upon reasonable request.

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
