# Peer review of "Advanced Skeletal Ossification Is Associated with Genetic Variants in Chronologically Young Beef Heifers"

_genes, 2023, doi:10.3390/genes14081629_

Round 1
Reviewer 1 Report
The article is interesting and innovative, providing SNPs associated with advanced skeletal ossification phenotypes in chronologically young beef heifers, and also validated previously reported variants. The results are meaningful, and will contribute to predicting genetic propensity for advanced skeletal ossification. The experiments were well designed, and the work load is very big. The results obtained were discussed efficiently. Furthermore, the article is well written. I am not proposing any improvement.
Reviewer 2 Report
The authors investigated the SNPs in candidate genes and found several genetic variants that may be associated with skeletal maturity. Several comments should be taken into consideration:
1. In this study, the constantly changing sample size is confusing. the constantly changing number of samples makes readers confused. In addition, it is difficult to understand the logical relationship between 793, 768, 634, 466, and 349 and the number of samples for each of the three groups correspondingly.
2. The current manuscript does not have evidence of the logistic regression and association analysis tests using Additive, Dominant, and Recessive models.
3. In the second paragraph of part 2.4, Supplemental Table 2 should be shown somewhere before “above for 13 candidate regions”. Why selected these regions as candidate regions?
4. ESR1 gene is located on chr10 in tables, mainly descriptions in the main text showed it is on chr9, except line6 in part 3.2.
Reviewer 3 Report
Dear Authors,
The proposed manuscript Advanced Skeletal Ossification is Associated with Genetic Variants in Chronologically Young Beef Heifers is interesting and scientifically sound.
In the interest of clarity of the manuscript, I would suggest that:
In Chapter 2. Materials and Methods (2.1. Sample Collection), you should list the genotypes that were included in the study (breeds, crossbreds, or something else).
In Chapter 2. Materials and Methods (2.6. Genetic Association Analyses), you should describe "Allelic, Basic, Genotypic, Additive, Dominant and Recessive models" in more detail so that the reader can understand how the assessment of association between SNPs and ossification was made
In Chapter 4. Discussion (4.1. SNP Validation; page 8), correct the sentence "The previously reported SNP within ALPL by Colacchio et al. (20) was validated in this study."
Regards,
Reviewer 4 Report
This study shows the identification of SNPs related to skeletal ossification, it is a complete work according to its resources. It would be interesting to study the relationship of these SNPs with reproductive aspects at an early age of the heifers, or probably if they have a long-term effect (maybe discuss this part a bit more).
Round 2
Reviewer 2 Report
All comments have been addressed, and I have no more comments.